# Vision-based Uneven BEV Representation Learning with Polar Rasterization and Surface Estimation

Zhi Liu [1,3] *    Shaoyu Chen [2,3] *    Xiaojie Guo [1†]    Xinggang Wang [2]
Tianheng Cheng [2,3]    Hongmei Zhu [3]    Qian Zhang [3]    Wenyu Liu [2]    Yi Zhang [1]

[1] Tianjin University    [2] Huazhong University of Science & Technology    [3] Horizon Robotics

**Abstract:** In this work, we propose PolarBEV for vision-based uneven BEV representation learning. To adapt to the foreshortening effect of camera imaging, we rasterize the BEV space both angularly and radially, and introduce polar embedding decomposition to model the associations among polar grids. Polar grids are rearranged to an array-like regular representation for efficient processing. To determine the 2D-to-3D correspondence, we iteratively update the BEV surface based on a hypothetical plane, and adopt height-based feature transformation. PolarBEV keeps real-time inference speed on a single 2080Ti GPU, and outperforms other methods for both BEV semantic segmentation and BEV instance segmentation. Thorough ablations are presented to validate the design. The code has been released for facilitating further research at https://github.com/SuperZ-Liu/PolarBEV.

**Keywords:** Polar Rasterization, Iterative Surface Estimation, BEV Segmentation

## 1 Introduction

Bird's Eye View (BEV) representation [1, 2, 3, 4, 5, 6, 7] is of great practical value for environmental perception in autonomous driving. Especially for vision-based system, BEV implicitly and elegantly aggregates multi-view information into a unified representation, avoiding time-consuming post processing for multi-view fusion.

This work proposes PolarBEV for vision-based uneven BEV representation learning. We rasterize the BEV space both angularly and radially, making BEV grids densely distributed near the ego-vehicle and sparsely distributed far from the ego-vehicle, *i.e.*, distance-dependent uneven grid distribution. Recent works [2, 8, 9] rasterize the BEV space along the cartesian axes and get evenly distributed rectangular grids. Such rectangular BEV representation is straightforward, but polar BEV representation makes more sense. First, for a self-driving car, the concerned perception region is centered at the ego vehicle. Surrounding perception results are more important than distant ones for avoiding traffic accidents. Thus, higher resolution in surrounding areas is expected. Second, for even BEV representation, long-range BEV space (*e.g.*, $100m \times 100m$) requires a large number of BEV grids and high computational budget. Polar rasterization enables long-tailed uneven grid distribution, which can be flexibly adjusted to cover large BEV space with limited computation cost.

We assign angle-specific and radius-specific embeddings to each polar grid according to its 3D position. Because of the foreshortening effects of camera imaging, object's scale in image varies a lot when the distance to camera changes. Polar grids at the same distance correspond to the same scale. Grids at the same angle correspond to the same camera view. With angle-specific and radius-specific embeddings, we model the associations among grids to enhance the BEV representation.

We propose iterative surface estimation for effective and efficient BEV representation learning. Previous methods [2, 9] usually predict pixel-wise depth distribution and broadcast pixel features to BEV space. Differently, we first set a hypothetical BEV surface and iteratively update the height of each polar grid to adjust the 2D-to-3D correspondence between image pixels and BEV grids. Height

6th Conference on Robot Learning (CoRL 2022), Auckland, New Zealand.

is much easier to be estimated than depth. The iterative refinement process leads to more precise 2D-to-3D feature transformation and better BEV representation.

PolarBEV achieves real-time inference speed (25 FPS on a 2080Ti GPU), and significantly outperforms counterparts for both BEV semantic segmentation and BEV instance segmentation. Given the high efficiency and strong performance, PolarBEV can be integrated into autopilot system for online environmental perception.

## 2   Related Work

Recently, many large multi-sensor datasets [10, 11, 12] made it possible to directly supervise models by projecting 3D annotations onto the ground plane to generate BEV labels. The key to the problem is how to model the transformation from image view to Bird's Eye View, which is inherently an ill-posed problem. A straightforward method is to assume the world is flat and transform the image to BEV map through Inverse Perspective Mapping (IPM) [1, 13, 14]. Though this approach works in some cases, it often introduces artifacts to objects that lie above the ground plane.

In order to achieve better results, other methods [15, 16, 17] explicitly estimate depth to lift objects into BEV. And OFT [18] maps image-based features into an orthographic 3D space with the aid of camera parameters. A potential performance bottleneck of this method is that the contribution of each pixel feature is independent of objects depth at that pixel. Instead of copying each pixel feature along camera ray, Lift-Splat [9] learns a depth distribution for each pixel. Recently, FIERY [2] has extended Lift-Splat [9] further to use multi-timestamp observations for motion forecasting. Different from these methods, we adopt BEV surface estimation instead of depth distribution to determine the correspondence between image and BEV.

Another technical route directly predicts BEV outputs from input images. CVT [8] encodes the camera parameters into positional embeddings to model the geometric structure of the scene implicitly. Different from CVT [8] using global attention to update each query, GKT [7] leverages the geometric priors to guide the transformer to focus on discriminative regions. VED [6] predicts a semantic occupancy grid directly from the front-view image with a variational encoder-decoder network. VPN [19] proposes a fully-connected view relation module to predict the semantic BEV map from multiple views. PON [20] further advances fully-connected layer for each column to translate features from image space to BEV space. Instead of using fully connected layers, TIM [21] models the relation of image columns and BEV polar rays with cross-attention. Based on TIM [21], the work [22] further employs a graph network to spatially reason about an object within the context of other objects. BEVFormer [23] predefines a set of uniformly distributed height anchors and projects these anchor points to image to get features.

In 3D domain, [24, 25] divide the 3D space into polar grids for point cloud segmentation, in order to adapt to the long-tailed distribution of LiDAR points. [26] leverages the radial symmetry to normalize point cloud along the radial direction. [27] introduces polar parametrization for 3D detection to establish explicit associations between image patterns and prediction targets. Differently, considering the foreshortening effects of camera imaging and the characteristics of BEV perception, we adopt polar rasterization for vision-based BEV representation learning.

## 3   Method

### 3.1   Overview

The framework of PolarBEV is presented in Fig. 1. Taking multi-view images as input, we first extract image features with shared CNN backbone. We rasterize the BEV space along the polar coordinates and rearrange the polar grids to array-like regular representation. Then, we iteratively update the BEV surface with grid-wise height estimation, and transform 2D features to BEV features based on the estimated height and camera's calibrated parameters. BEV surface updating and feature transformation are repeated in a cascade manner for several times. And various heads follow the final BEV representation to perform BEV perception. Detailed designs are presented below.

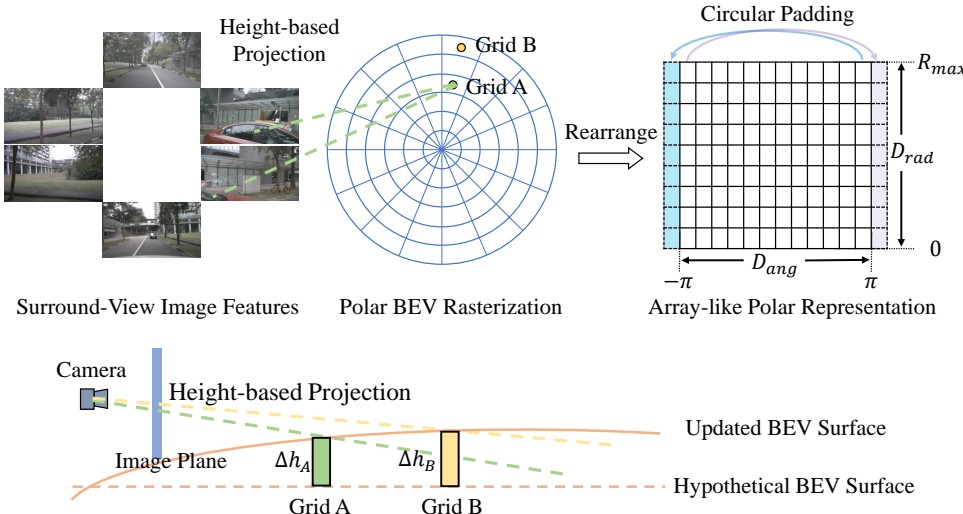

Figure 1: Illustration of PolarBEV. Top: multi-view images are sent into the shared CNN backbone for feature extraction. BEV space is rasterized along the polar coordinates. Polar girds are rearranged to array-like regular representation. Bottom: we iteratively update the BEV surface for precise height-based projection.

### 3.2 Polar BEV Rasterization and Rearrangement

The concerned BEV space is centered at the ego-vehicle with the radius $R_{max}$ and 360° FoV (field of view) coverage. As shown in Fig. 1, we rasterize the BEV space both angularly and radially. Radially, we evenly divide $[0, R_{max}]$ into $D_{rad}$ segments ($R_{max}$ and $D_{rad}$ respectively denote radius maximum and radial resolution). Angularly, we evenly divide the 360° into $D_{ang}$ segments ($D_{ang}$ denotes the angular resolution). After rasterization, for efficiently processing the polar representation, we rearrange polar grids along the angular and radial dimensions and get array-like regular representation with shape $D_{rad} \times D_{ang}$. The rearranged representation is hardware-friendly but can not be processed by conventional convolution operation. As shown in Fig. 1, in the angular dimension, $-\pi$ and $\pi$ correspond to the the same angle but are separated in the array-like regular representation. Alternatively, we adopt ring convolution [25] with circular padding to process the rearranged representation. Specifically, we first circularly pad in the angular dimension, and then adopt conventional convolution with 0 padding in the radial dimension. The ring convolution is followed by batch normalization and ReLU activation layers.

### 3.3 Polar Embedding Decomposition

For each polar grid with radial distance $r$ and angle $\theta$, we predefine a learnable query embedding $q$ and decompose $q$ into two components, *i.e.*, radius-specific query $q_{rad}^{(r)}$ and angle-specific query $q_{ang}^{(\theta)}$, which is formulated as,

$$q = q_{rad}^{(r)} + q_{ang}^{(\theta)}. \tag{1}$$

For the polar representation, polar grids at the same distance correspond to the same scale, and grids at the same angle correspond to the same camera view. With angle-specific and radius-specific embeddings, we model the associations among grids to enhance the BEV representation. Ablation experiments are presented in Sec. 4.3 to validate the effectiveness of embedding decomposition.

### 3.4 Iterative Surface Estimation and 2D-to-3D Feature Transformation

We perform 2D-to-3D feature transformation based on a height-based projection mechanism. To make sure the correspondence between image and BEV, we first set a hypothetical BEV surface with height $h_{hypo}$ and then iteratively update the height of each grid $i$ based on its query embedding

$q_i$. This can be formulated as follows:

$$h_i^t = \Theta(q_i^{t-1}) + h_i^{t-1}, \tag{2}$$

where $\Theta$ is a MLP layer, $t$ is the iteration index and $h_i^0$ is the hypothetical height $h_{hypo}$. Then we normalize $h_i$ (superscript $^t$ is omitted for clarity) to the range of $[0, 1]$ with sigmoid function $\sigma$ and further scale up the value to the range of $[Z_{inf}, Z_{sup}]$, i.e.,

$$z_i = \sigma(h_i) \times (Z_{sup} - Z_{inf}) + Z_{inf}, \tag{3}$$

where $Z_{sup}$ and $Z_{inf}$ are the predefined upper and lower bounds of height respectively. Each grid $i$ corresponds to a polar coordinate $p_i = (r_i, \theta_i)$ (the position of the grid's center point). We first transform each polar coordinate $p_i$ to cartesian coordinate $(x_i, y_i)$ in the following manner:

$$x_i = r_i \times \cos\theta_i, \quad y_i = r_i \times \sin\theta_i. \tag{4}$$

Then we concatenate the cartesian coordinate $(x_i, y_i)$ with its corresponding height $z_i$ to construct the homogeneous coordinate $w_i$, i.e.,

$$w_i = (x_i, y_i) \oplus z_i \oplus 1, \tag{5}$$

where $\oplus$ denotes the concatenation operation. Then we project $w_i$ to the image plane with camera's intrinsics $I \in \mathbb{R}^{K \times 3 \times 3}$ and extrinsics $E \in \mathbb{R}^{K \times 3 \times 4}$:

$$u_{i,n} = I_n \cdot E_n \cdot w_i^T, \tag{6}$$

where $u_{i,n}$ is the projected image point on the $n$th view of BEV grid $i$, and $K$ represents the total image views. Finally, we transform features from the image view to the Bird's Eye View with the projected coordinates. This can be formulated as:

$$f_i = \sum_{n=1}^{K} M_n \cdot F_{i,n}, \tag{7}$$

where $F_{i,n}$ is the sampled image feature corresponding to the projected image point $u_{i,n}$, and $M_n$ is a binary mask for masking out projected points which exceed the image boundary. After that, $f_i$ is sent into a MLP layer and its output is added to the previous query for query update.

The surface estimation and 2D-to-3D feature transformation are conducted for several times and the polar representation is iteratively updated.

### 3.5 Segmentation Head

We follow the head design of FIERY [2]. We adopt a small encoder-decoder network to further refine BEV features. Three branches follow the network, respectively for predicting segmentation score, offset and centerness. The loss settings are also the same with FIERY [2]. Polar predictions are mapped into rectangular predictions for loss calculation based on the rectangular ground truths.

## 4 Experiments

### 4.1 Experimental Settings

**Datasets** NuScenes [10] dataset is a large-scale autonomous driving dataset which was collected over a variety of weathers and traffic conditions. This dataset contains 1000 scenes and each scene lasts 20 seconds. The captured RGB images are from six cameras covering a full of $360^{\circ}$ around the ego-vehicle. Each camera has calibrated intrinsics and extrinsics at every timestamp. Following common practice [2, 8], we project the 3D box annotations of vehicles onto the ground plane to get ground truth labels.

**Architecture** We adopt the pre-trained EfficientNet-B4 [28] as backbone to extract multi-view image features. Features of stride 8 are taken as the input for feature transformation. The initial polar BEV feature is a combination of angular queries and radial queries with the shape of $400 \times 100 \times 64$ for representing a certain BEV area (setting 1 or setting 2) centered at the ego-vehicle. We set the radius maximum $R_{max}$ as the distance from the farthest point of BEV area to the center of ego-vehicle. The small encoder-decoder network contains first four layers of ResNet-18 [29], and all the convolution layers with kernel size greater than 1 are changed into ring convolutions to adapt to the polar representation.

| Method | Setting 1(IoU%) | Setting 2(IoU%) | # Params(M) | FPS |
|---|---|---|---|---|
| PON [20] | 24.7 | - | 38 | 30 |
| VPN [19] | 25.5 | - | 18 | - |
| STA [33] | 36.0 | - | - | - |
| Lift-Splat [9] | - | 32.1 | 14 | 25 |
| FIERY Static [2] | 37.7 | 35.8 | 7.4 | 8 |
| Ours (224×480) | **41.5** | **37.6** | 7.4 | 25 |
| Ours (448×960) | **45.4** | **41.2** | 7.4 | 10 |

Table 1: Comparison of vehicle BEV semantic segmentation on nuScenes [10] **without** masking invisible vehicles. Setting 1 refers to the $100m \times 50m$ at $25cm$ resolution. Setting 2 refers to the $100m \times 100m$ at $50cm$ resolution. We report Intersection-Over-Union (IoU) for evaluating vehicle BEV semantic segmentation performance. The iteration of feature transformation is set to 2 for both input resolution, (224×480) and (448×960) in this table. Our method (224×480 input) significantly outperforms counterparts while achieving real-time speed (25 FPS).

| Method | Setting 1(IoU%) | Setting 2(IoU%) | # Params(M) | FPS |
|---|---|---|---|---|
| FIERY Static [2] | 42.7 | 39.8 | 7.4 | 8 |
| CVT [8] | 37.5 | 36.0 | 5 | 35 |
| Ours (224×480) | **44.3** | **41.3** | 7.4 | 25 |
| Ours (448×960) | **48.4** | **45.6** | 7.4 | 10 |

Table 2: Comparison of vehicle BEV semantic segmentation on nuScenes [10] **with** masking invisible vehicles. The iteration of feature transformation is set to 2 for both input resolution, $(224 \times 480)$ and $(448 \times 960)$ in this table. Our method (224×480 input) achieves much higher IoU than FIERY Static [2] and CVT [8] while achieving real-time speed (25 FPS).

**Training** All our networks are implemented with Pytorch [30] and Pytorch Lightning. The input images are first resized and randomly cropped into a special resolution before being fed into the networks. The AdamW [31] optimizer is used for training the networks, with weight decay of $1e^{-7}$. We apply the one-cycle learning rate scheduler [32] to adjust the learning rate. All networks are trained on 8 GPUs with a batch size of 16.

**Evaluation** Two settings are used for evaluating vehicle segmentation map. Setting 1 [20] perceives a $100m \times 50m$ area around the ego-vehicle with $0.25m$ resolution, while the perceptual range of setting 2 [9] is $100m \times 100m$ with $0.5m$ resolution. These two settings serve as the main comparisons to prior works. We use setting 2 for all the ablation studies. The Intersection-Over-Union (IoU) metric is used for evaluating vehicle BEV segmentation performance for both settings. We also report Panoptic Quality (PQ) metric for evaluating vehicle BEV instance segmentation performance, following previous work [2]. Additionally, we report the inference speeds measured on a single 2080Ti GPU. The number of iterations and input resolution for PolarBEV are set to 3 and $224 \times 480$ respectively for most experiments unless specified otherwise. Results of ablation studies are produced with masking invisible vehicles.

### 4.2 Performance Comparison

To validate the effectiveness and efficiency of the proposed method, we compare PolarBEV with competitive approaches on both BEV semantic segmentation and BEV instance segmentation. All the methods involved in comparison only use single frames as input without temporal information.

**BEV Semantic Segmentation** For fair comparison, in Tab. 1 and Tab. 2, we respectively compare results without and with masking invisible vehicles. Tab. 1 compares results without masking invisible vehicles. Our model with 2 iterations outperforms counterparts by a significant margin in both settings, especially for $448 \times 960$ input resolution. Even compared with the best prior method FIERY Static [2], our method ($224 \times 480$ input) is still 3.8 points higher in setting 1 and performs $3 \times$ faster inference with the same parameters. When the input resolution is increased to $448 \times 960$, our model

|  | Setting 1 | | | Setting 2 | | | FPS |
|---|---|---|---|---|---|---|---|
|  | RQ% | SQ% | PQ% | RQ% | SQ % | PQ% |  |
| FIERY Static [2] | 49.5 | 71.7 | 35.5 | 47.3 | 71.1 | 33.6 | 8 |
| Ours | **52.3** | **72.0** | **37.7** | **50.2** | **71.4** | **35.9** | **25** |

Table 3: Comparison of vehicle BEV instance segmentation on nuScenes [10] with masking invisible vehicles. We report Recognition Quality (RQ), Segmentation Quality (SQ) and Panoptic Quality (PQ) for evaluating vehicle BEV instance segmentation performance. Our method significantly outperforms FIERY Static [2] in both performance and speed.

| Grid Distribution | IoU% | PQ% | FPS |
|---|---|---|---|
| Rectangular | 40.83 | 35.45 | 22 |
| Polar | **41.36** | **35.78** | 22 |

Table 4: Ablation study about polar and rectangular grid distribution.

| Feature Transformation | IoU% | PQ% | FPS |
|---|---|---|---|
| Depth-based | 39.81 | 33.62 | 8 |
| Height-based | **40.83** | **35.45** | 22 |

Table 5: Ablation study about depth-based and height-based feature transformation.

achieves further 7.7 points higher in setting 1 but still runs in a faster speed than FIERY Static. In Tab. 2, we compare PolarBEV with other methods with masking invisible vehicles. Because there is no official results of FIERY Static [2] in this setting, we reproduce it with the official code. As this table shows, our model ($224 \times 480$ input) with 2 iterations also achieves the best results in both settings and achieves real-time performance.

**BEV Instance Segmentation**  We also provide BEV instance segmentation comparison with masking invisible vehicles in Tab. 3. The results of FIERY Static [2] are reproduced with the official code. As shown in this table, our model with 2 iterations is more than two points higher than FIERY Static in both settings in terms of PQ. We also report Recognition Quality (RQ) and Segmentation Quality (SQ) in this table. As we can see from this table, the revenue mainly comes from RQ, meaning our model detects instances more accurately.

## 4.3 Ablation Study

**Polar *vs.* Rectangular Grid Distribution**  In Tab. 4, we present ablation experiments about polar and rectangular grid distribution. For fair comparison, both results have the same number of BEV grids, $200 \times 200$ for rectangular grid distribution and $400 \times 100$ for polar grid distribution. We keep the network structure the same. As shown in Tab. 4, polar grid distribution achieves an improvement of 0.53 IoU and 0.33 PQ with no degradation in inference speed.

**Height-based *vs.* Depth-based Feature Transformation**  In Tab. 5, we provide ablation experiments about feature transformation. FIERY Static [2] adopts depth-based transformation, thus we directly take it as baseline for comparison. To get the height-based results, based on FIERY Static, we change the feature transformation manner to the one introduced in Sec 3.4. Tab. 5 shows that height-based results surpass depth-based results a lot in both IoU and PQ with nearly $3\times$ inference speed, validating the advantage of height-based transformation.

The advantage mainly comes from the following points. The numerical range of depth is $[0, +\infty)$ for each pixel, which is hard for the network to estimate a reasonable value in such a huge space. While

| Ring Convolution | PED ($q_{ang}, q_{rad}$) | IoU% | PQ% |
|---|---|---|---|
|  |  | 40.48 | 34.39 |
| ✓ |  | 40.81 | 35.42 |
| ✓ | ✓ | **41.36** | **35.78** |

Table 6: Ablation study about ring convolution and embedding decomposition. PED denotes Polar Embedding Decomposition.

| Angular | Radial | IoU% | PQ% | # Params(M) |
|---|---|---|---|---|
| 50 | 100 | 37.43 | 26.63 | 7.5 |
| 100 | 100 | 39.53 | 31.77 | 7.5 |
| 200 | 100 | 40.46 | 34.63 | 7.5 |
| 400 | 100 | 41.36 | 35.78 | 7.5 |
| 600 | 100 | 41.40 | 35.93 | 7.6 |
| 400 | 50 | 40.34 | 34.60 | 7.5 |
| 400 | 142 | **41.48** | **36.13** | 7.5 |

Table 7: Ablation study about various resolutions of polar BEV representation. The shape of polar BEV representation is $D_{ang} \times D_{rad}$.

| t | IoU% | PQ% | #Params(M) | FPS |
|---|---|---|---|---|
| 1 | 40.73 | 34.59 | 7.3 | **27** |
| 2 | 41.30 | 35.86 | 7.4 | 25 |
| 3 | 41.36 | 35.78 | 7.5 | 22 |
| 6 | **41.39** | **35.96** | 7.8 | 17 |

Table 8: Ablation study about iterations of the 2D-to-3D feature transformation.

the numerical range of height for BEV is quite small. Height-based transformation corresponds to less prediction error. As for the speed, depth-based methods (such as FIERY Static) broadcast pixel features to BEV space along the depth dimension and sum all the 3D features along the vertical dimension. These operations are time-consuming. While height-based transformation is lightweight, resulting in less latency.

**Ring Convolution and Polar Embedding Decomposition**    In Tab. 6, we ablate on the ring convolution and the proposed polar embedding decomposition. The baseline result achieves $40.48\%$ IoU and $34.39\%$ PQ. After adopting ring convolution, we observe an improvement of $0.33$ in IoU and $1.03$ in PQ respectively. In order to model the associations among polar grids, we decompose grid embeddings $q$ into angle-specific ones $q_{ang}$ and radius-specific ones $q_{rad}$. As the last row shows, embedding decomposition further brings an improvement of $0.55$ IoU and $0.36$ PQ.

**Polar BEV Resolution**    To verify how the resolution of polar BEV representation affects model performance, we performe ablation experiments on the angular resolution and the radial resolution respectively. As shown in the upper of Tab. 7, the model performance becomes better with the increase of angular resolution and reaches saturation at $400$. The same rule can be found in radial resolution, as the radial resolution increases, the results become better. Although increasing the angular resolution or the radial resolution brings benefits and barely increases model parameters, the memory occupation is proportional to the angular or radial resolution. Considering both the performance and memory occupation, we choose the resolution of $400 \times 100$ as the default setting.

**The Number of Iteration**    We also conduct ablation study about the number of BEV surface estimation iterations in Tab. 8. As this table shows, our model with only 1 iteration gets $40.73\%$ IoU and $34.59\%$ PQ, which has already exceeded all previous methods by a significant margin but with a real-time inference speed. When the number of iteration is 2, the method achieves higher performance in both IoU and PQ metrics and still keeps real-time speed. After adding the number of iteration to 3 or 6, the performance seems to reach saturation, while the FPS degrades a lot when the number of iterations is 6.

## 4.4   Qualitative Results

Fig. 2 shows several qualitative results on various scenes. We give the six camera views and the predicted segmentation results and instance segmentation results along with the ground truth in each row. As can be seen from this figure, PolarBEV can accurately segment vehicles in a variety of complex scenes.

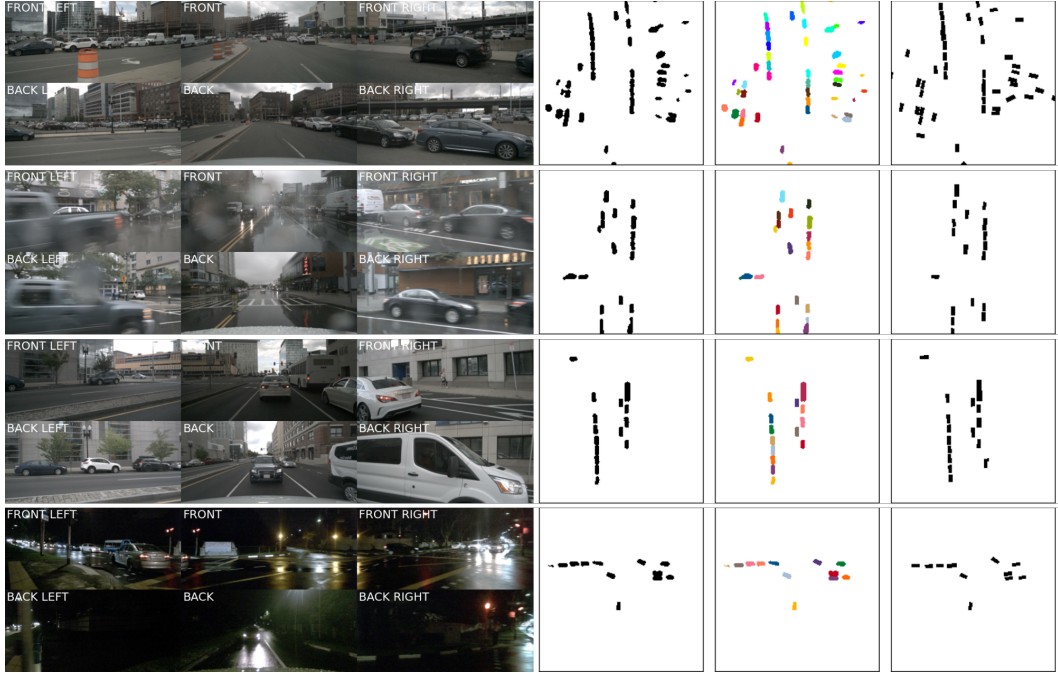

Figure 2: Qualitative results on various scenes. Left shows the six camera views. Right shows our semantic segmentation results, instance segmentation results and the ground truth in turn.

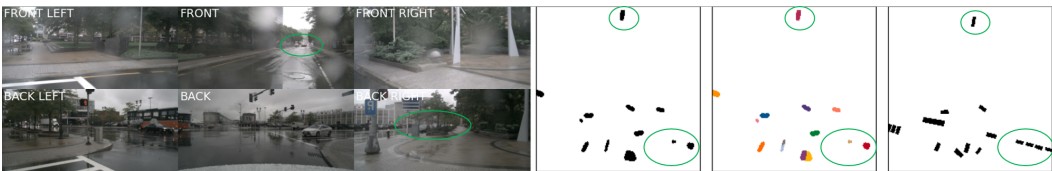

Figure 3: Failure examples on nuScenes [10]. Left shows the six camera views. Right shows our semantic segmentation results, instance segmentation results and the ground truth in turn.

## 4.5 Limitations

We present some failure examples in Fig. 3. As shown in the upper of this figure, it is hard for PolarBEV to predict the exact location when vehicle is far away from the ego-vehicle. As the bottom of this figure shows, PolarBEV fails when the height of vehicle is hard to estimate because of blocking. In order to achieve better results in the future, we may consider using temporal information to calibrate the failures.

## 5 Conclusion

We propose PolarBEV for vision-based uneven BEV representation learning in this work. PolarBEV rasterizes the BEV space both angularly and radially, making a distance-dependent uneven grid distribution. To model the associations among grids, we assign angle-specific and radius-specific embeddings to each polar grid. Different from previous depth based methods, we first set a hypothetical BEV surface and then iteratively update the height of each polar grid to adjust the 2D-to-3D correspondence between image pixels and BEV grids. Extensive experiments show PolarBEV is an alternative way for better and faster segmentation.

**Acknowledgments**

This work was supported by the National Natural Science Foundation of China under Grant no. 62072327.

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
