# OpenReview forum: "Vision-based Uneven BEV Representation Learning with Polar Rasterization and Surface Estimation"
_robot-learning.org/CoRL/2022/Conference — CoRL 2022 Poster_

### Official Review · Reviewer_SUoV · 2022-07-28

**Originality:** Fair
**Technical Quality:** Good
**Clarity Of Presentation:** Very Good
**Impact:** 3

**Recommendation:**

Weak Reject: I recommend rejecting the paper, but will not argue for my recommendation if the majority of other reviewers have a different opinion.

**Summary:**

This paper tackles vision-based BEV semantic segmentation for on-road driving. It improves the previous work, FIERY, in two aspects: (i) using polar grid instead of a rectangular grid, and (ii) estimating the height of each map location instead of estimating the depth of each image pixel. Experiments validate the design choice and show that both the polar representation and the height estimation improve the BEV segmentation results on the nuscenes dataset.

**Issues:**

* Please address the issues raised the previous section, namely i) why polar is better than rectangular and ii) the relation to existing work.
* Make the technical section (3.4) clearer.

**Quality Of The Limitations Section:**

Limitations are addressed clearly

**Reviewer Expertise:**

3: The reviewer is fairly confident that the evaluation is correct

**Robotics Focus:**

Highly relevant to robotics but no hardware experiments

**Strengths And Weaknesses:**

Strengths:

- The proposed ideas are intuitive, effective and easy to implement.
- The idea of iteratively estimating the height is interesting.
- Experiments are thorough with detailed ablation study.

Weaknesses:

The idea of using polar representation and height-based projection are not that new. There has been plenty of work of adopting polar representation for on-road driving (the authors have mentioned in the related work). Recently, there is also work that proposes height estimation for BEV segmentation [1]. While this work shows that the proposed methods are better than the baseline, I have several concerns of the proposed approach.

[W1] Although the empirical results show that the polar representation is better, it is not clear why. Polar representation would have higher resolution close to the center, but its resolution at the edge of the map is lower than the rectangular representation. I think there is a tradeoff, and I wonder how the polar representation would perform as a function of the distance to the center. Is it always better? Or is it better near the center, and worse when far from the center than the rectangular representation? Does the dataset have a bias like this? It would be great if the authors could elaborate on this.

[W2] While adding height estimation seems effective, it is unclear how well the height estimation works. Does it predict reasonable height values? Currently, it shows that the height estimation helps, but it is unclear if it is helping in the right way, i.e. predicting reasonable height values.

[W3] The height estimation only estimates a single height value for each map cell. This discards a lot of the information from images. Does this have any potential drawbacks? A similar work [1] alleviates this issue by maintaining multiple height values in each map cell. It would be great if the authors could clarify the difference between this work and [1].

[W4] The technical section could be made clearer.
- In Sec 3.4, it is unclear how the sampled features $f_i$ be used to produce the query embedding $q$ in the next iteration.
- I wonder if the comparison with previous work FIERY is fair. Are you using the same network architecture and parameter settings?

[1] BEVFormer: Learning Bird's-Eye-View Representation from Multi-Camera Images via Spatiotemporal Transformers arXiv preprint arXiv:2203.17270


**Summary Of Recommendation:**

The paper contains some interesting ideas and is enjoyable to read. While the ideas are not entirely novel, they are applied to a slightly different task or in a different way, and detailed experiments have shown some advantages over existing methods. However, I would like to see some clarifications about where the performance gain comes from, and the relation to related work. It would be great if the authors could provide some additional details about the issues raised.

---

> ### Author Response · Authors · 2022-08-28
> **Response to Reviewer SUoV (1 / 2)**
>
> Thanks for the helpful comments to our work. Following are our detailed responses to your concerns.
>
> $\color{red}Question: $ The idea of using polar representation and height-based projection are not that new. There has been plenty of work of adopting polar representation for on-road driving (the authors have mentioned in the related work). Recently, there is also work that proposes height estimation for BEV segmentation [1].
>
> $\color{blue}Response: $ First of all, we would like to respectfully argure that the idea of using polar representation and height-based projection is novel and new (as also recognized by the other two reviewers).
>
> For the polar representation, there has been few works adopting polar representation for on-road driving. The related works [2, 3]  belong to the domain of point cloud parsing, but our method is designed for vision-based BEV representation learning.  The gap between point cloud and surround-view images is non-trivial. [2, 3] and our method only share the same conception of 'polar', but the pipelines and detailed implementation techniques are totally different (e.g., the 2D-to-3D transformation).
>
> For the height-based projection, we argue that BEVFormer [1] does not propose height estimation and is not height-based. BEVFormer predefines a set of fixed heights (-4m, -2m, 0m, 2m)  and projects BEV cells to 2D images with these fixed heights to get fixed reference points in 2D images. BEVFormer dynamically samples features with Deformable Convolution on 2D images. But the heights are not learnable and are always the same for all BEV cells. Differently, our method iteratively updates the height to adjust the 2D-to-3D correspondence between image pixels and BEV cells. The heights are learnable and dynamic. For different BEV cells and different driving scenes, the estimated heights are different. Thus, BEVFormer is quite different from our method.
>
> [2] H. Zhou, et. al. Cylinder3d: An effective 3d framework for driving-scene lidar semantic segmentation. arXiv preprint arXiv:2008.01550, 2020.
> [3] Y. Zhang, et. al. Polarnet: An improved grid representation for online lidar point clouds semantic segmentation. CVPR, 2020.
>
> $\color{red}Question: $ I would like to see some clarifications about where the performance gain comes from.
>
> $\color{blue}Response: $ Please refer to Tab. 4 and Tab. 5 in the original paper. We have provided ablation experiments to respectively validate the design of polar parameterization and height-based transformation, both of which do bring improvement in terms of inference speed and performance.
>
> $\color{red}Question: $ Although the empirical results show that the polar representation is better, it is not clear why. Polar representation would have higher resolution close to the center, but its resolution at the edge of the map is lower than the rectangular representation. I think there is a tradeoff, and I wonder how the polar representation would perform as a function of the distance to the center. Is it always better? Or is it better near the center, and worse when far from the center than the rectangular representation? Does the dataset have a bias like this? It would be great if the authors could elaborate on this.
>
> $\color{blue}Response: $ To response to your concern, in Tab.1, we compare the performance of  FIERY Static (rectangular representation) and our polarBEV (polar representation) at different radial distances, based on the same network architecture and parameter settings. The results show that the performance gradually drops with the increase of distance, fitting with the common sense. And polar representation achieves better results than rectangular representation at most radial distances, except that it's slightly worse nearby the ego-vehicle (0-10m). We think it's because polar grids are too dense in the surrounding region, resulting in limited receptive field and insufficient context. It could be improved by enhancing the context information.
> | Radial Distance | 0-10m | 10-20m | 20-30m | 30-40m | 40-50m |
> | ------ | ------ | ------ | ----- | ------ | ----- |
> | IoU of FIERY Static (rectangular)  |      73.2   |    54.5      |     39.7    |       26.6    |  16.4 |
> | IoU of PolarBEV (polar)       |   70.9    |    56.1       |    41.6       |    29.2        |  18.2 |
>  Table 1: Performance comparison of FIERY Static and PolarBEV at different radial distances.
>
> And the superior performance of our method does not result from the bias of dataset. To validate this, we further conduct experiments on Lyft [4] dataset and compare with FIERY Static. As shown in Tab. 2, under the same experiment settings, our method significantly outperforms FIERY Static in both IoU and FPS, consistent with the results on nuScenes dataset.
> | Method |IoU| FPS|
> | ------ | ------ | ------ |
> | FIERY Static| 42.25 | 8 |
> | Ours | 44.69 | 22 |
> Table 2: Lyft dataset results.
>
> [4] R. Kesten et. al. Lyft level 5 perception dataset. 2020, 2019.

---

> ### Author Response · Authors · 2022-08-28
> **Response to Reviewer SUoV (2 / 2)**
>
> $\color{red}Question: $ While adding height estimation seems effective, it is unclear how well the height estimation works. Does it predict reasonable height values? Currently, it shows that the height estimation helps, but it is unclear if it is helping in the right way, i.e. predicting reasonable height values.
>
> $\color{blue}Response: $ Since the height is not explicitly supervised by ground-truth annotations,  it's qute difficult to predicting totally exact height. But with the iterative update, the predicted height is more and more accurate and reasonable. We calculate the height error of different iterations to prove it.  As shown in Tab.3, as the estimation iterates, the height error gradually decreases, which corroborates the effectiveness of the iterative height update mechanism.
> | Iteration |1 | 2 | 3 |
> | ------ | ------ | ------ | ----- |
> |Height error| 1.7252m  | 1.5630m  | 1.4351m|
>  Table 3:  Height error of different iterations.
>
> $\color{red}Question: $ The height estimation only estimates a single height value for each map cell. This discards a lot of the information from images. Does this have any potential drawbacks? A similar work [1] alleviates this issue by maintaining multiple height values in each map cell. It would be great if the authors could clarify the difference between this work and [1].
>
> $\color{blue}Response: $ As we have discussed above, BEVFormer[1] is not  height-based and is quite different from our method. BEVFormer predefines a set of fixed heights (-4m, -2m, 0m, 2m)  and projects BEV cells to 2D images with these fixed heights to obtain fixed reference points in a 2D image. BEVFormer dynamically samples features with Deformable Convolution on 2D images. But the heights are not learnable and are always the same for all BEV cells.  Differently, our method iteratively updates  the heights  to adjust the 2D-to-3D correspondence between image pixels and BEV cells. Again, we emphasize that the heights are learnable and dynamic. For different BEV cells and different driving scenes, the estimated heights are different. Thus, BEVFormer and our method differs significantly  in 2D-to-3D transformation. Besides, BEVFormrer is based rectangular representation, quite different from our proposed polar representation.
>
> Estimating several height values bring much more computation costs. Our method aims at real-time environmental perception, which is required by autopilot system. Thus we only adopt one height for the performance-speed trade-off.
>
> $\color{red}Question: $ In Sec 3.4, it is unclear how the sampled features $f_i$ be used to produce the query embedding $q$ in the next iteration.
>
> $\color{blue}Response: $ The implementation is straightforward. We send  $f_i$ into a MLP layer and the output of MLP is added to the previous query for query update. Thanks for the advice and we have revised the Sec 3.4 to make this detail clear.
>
> $\color{red}Question: $ I wonder if the comparison with previous work FIERY is fair. Are you using the same network architecture and parameter settings?
>
> $\color{blue}Response: $ As for network architecture, we do adopt the SAME backbone (EfficientNet-B4) and head design with FIERY. The only difference is the proposed depth-based projection and polar representation.
>
> As for parameter settings,  we adopt the same parameter settings in all ablation experiments for fair comparison. For reporting the main results, our parameter settings are based on both FIERY and CVT.
>
> To further respond to the concern, we conduct addtional experiments to compare FIERY Static and our method based on the totally same parameter settings.  As shown  in Tab.4, the results show that our method still achieves higher performance and FPS than FIERY Static under the totally same configuration.
>
> | Method |IoU| FPS|
> | ------ | ------ | ------ |
> | FIERY Static| 40.48 | 8 |
> | Ours | 41.36 | 22 |
> Table 4: Comparsion between FIERY Static and our method based on the totally same configuration.

---

### Official Review · Reviewer_iJAs · 2022-08-01

**Originality:** Good
**Technical Quality:** Good
**Clarity Of Presentation:** Good
**Impact:** 2

**Recommendation:**

Weak Accept: I recommend accepting the paper, but will not argue for my recommendation if the majority of other reviewers have a different opinion.

**Summary:**

The paper proposes a method for projecting features from 2D images into a polar bird eye's view grid. This grid is then used for estimating a 2.5D height surface of what is around the cameras. Features are then re-projected into an array-like polar representation and fed into an encoder-decoder based on FIERY [2].

**Issues:**

## Major comments
* The notations for variables have several typos, which makes it confusing to follow, e.g., line 111 and 112, are {$z_{inf}, Z_{inf}$} and {$z_{sup}, Z_{sup}$} the same things? Same goes for line 100, and equation (1) with $p_{ang}$ and $q_{ang}$.
* I do not understand how adding $q_{rad}$ and $q_{ang}$ allows forming a meaningful query system. How do you distinguish {$q_{ang} = 10^\circ, q_{rad} = 1m$} from {$q_{ang} = 1^\circ, q_{rad} = 10m$} (obviously things might be quantified differently but I think I missed something).
* The methodology is hard to follow, I would encourage the author to clarify the distinction between the two types of projections (the first one with 3.2 and the second one in 3.4) by making the headings consistent with figure 1. Also, specify explicitly before equation (5) that you are doing the image re-projection.
* Clarify what is the variable t, is equation (2) solved as a sliding window between frames, or is wrt to cells with closer radius, or is solved for each frame as an iterative optimisation starting from h_hypo until convergence?

## Minor comments
* line 150, do you mean 30k epochs? How do you track overfitting?
* If setting 1 and 2 are used for evaluation, which settings do you use for training (Or how are the datasets divided between training and evaluation)?
* Line 168, you compare with FIERY static, and not FIERY
* Line 177, no need to redefine acronyms
* Is using a constant resolution for $D_{rad}$ the best approach?

**Quality Of The Limitations Section:**

Limitations are addressed clearly

**Reviewer Expertise:**

3: The reviewer is fairly confident that the evaluation is correct

**Robotics Focus:**

Sufficient demonstration on hardware

**Strengths And Weaknesses:**

The authors propose to use polar rasterization for images inputs. This is a novel approach that changes how the inputs of a neural network can be encoded. The method is well evaluated and show some improvements when compared to the state of the art.

The clarity of the method section needs to be improved as some of the descriptions are confusing and could interpreted in different ways.


**Summary Of Recommendation:**

I do think the paper proposes an interesting contribution with is novel and interesting. I do think the writing needs to be improved for a more clear, simpler, and more consistent methodology. That aside the experiments are pretty thorough with ablation studies of several of the parameters and comparison with state of the art.

---

> ### Author Response · Authors · 2022-08-28
> **Response to Reviewer iJAs**
>
> Thanks for recognizing the novelty and contribution of our work! We have improved the writing in the revised paper  following your suggestion. Below are our responses to other detailed concerns.
>
> $\color{red}Question: $ The notations for variables have several typos, which makes it confusing to follow, e.g., line 111 and 112, are \{$z_{inf}, Z_{inf}$\} and \{$z_{sup}, Z_{sup}$\} the same things? Same goes for line 100, and equation (1) with $p_{ang}$ and $q_{ang}$.
>
> $\color{blue}Response: $ Sorry for these typos. We have corrected these variables in the revised paper.
>
> $\color{red}Question: $ I do not understand how adding $q_{rad}$ and $q_{ang}$ allows forming a meaningful query system. How do you distinguish \{$q_{ang}=10^\circ,q_{rad}=1m$\} from \{$q_{ang}=1^\circ,q_{rad}=10m$\} (obviously things might be quantified differently but I think I missed something).
>
> $\color{blue}Response: $ All polar grids with the same radial distance $r$  share the same radius-specific query $q_{rad}^{(r)}$. All polar grids with the same angle $\theta$  share the same angle-specific query $q_{ang}^{(\theta)}$. For each polar grid, its query $q$ is composed of a radial component  $q_{rad}$  and an angular component $q_{ang}$.  The polar grid $(10^{\circ}, 1m)$   corresponds to $q_{ang}^{(10^{\circ})}$  and   $q_{rad}^{(1m)}$, while the polar grid  $(1^{\circ}, 10m)$ corresponds to $q_{ang}^{(1^{\circ})}$ and $q_{rad}^{(10m)}$.
>
> $\color{red}Question: $ The methodology is hard to follow, I would encourage the author to clarify the distinction between the two types of projections (the first one with 3.2 and the second one in 3.4) by making the headings consistent with figure 1. Also, specify explicitly before equation (5) that you are doing the image re-projection.
>
> $\color{blue}Response: $ Thanks for the kind advice. Sec.3.2 is about the polar distribution of BEV grids and Sec.3.4  is about the projection between polar grids and images. We have revised the methodology part for better readability. And we will release the code as soon as possible and readers can check our code to get more implementation details.
>
> $\color{red}Question: $ Clarify what is the variable $t$, is equation (2) solved as a sliding window between frames, or is wrt to cells with closer radius, or is solved for each frame as an iterative optimisation starting from $h_{hypo}$ until convergence?
>
> $\color{blue}Response: $ The variable $t$ is sovled for each frame as an iterative operation starting from $h_{hypo}$. It represents the iteration number of BEV surface updating and feature transformation. We have added explanations about variable $t$ in the revised paper to make it clear (Line 113).
>
> $\color{red}Question: $ line 150, do you mean 30k epochs? How do you track overfitting?
>
> $\color{blue}Response: $ It means $30k$ iterations for updating the model parameters. We train these mdoels with the batch size of $16$ and $30k$ iterations. The nuscenes dataset contains about $27k$ annotated samples and correspondingly the total epochs are about $17$. We have checked the loss, from which we observed that, the validation loss stably decreases in the training process,  and our model has little overfiting phenomenon.
>
> $\color{red}Question: $ If setting 1 and 2 are used for evaluation, which settings do you use for training (Or how are the datasets divided between training and evaluation)?
>
> $\color{blue}Response: $ We train the model twice respectively for setting 1 and setting 2. The training and evaluation are based on the consistent setting. Setting 1 and setting 2 mainly differ in BEV range and resolution. Both settings are adopted by previous works (like FIERY and CVT). We also take these settings for thorough comparison.
>
> $\color{red}Question: $ Line 168, you compare with FIERY static, and not FIERY.
>
> $\color{blue}Response: $ FIERY Static denotes a result reported by the FIERY method. In L168, we mean 'compared with FIERY method'. For avoiding ambiguity, we have changed it to FIERY Static in the revised paper.
>
> $\color{red}Question: $ Line 177, no need to redefine acronyms.
>
> $\color{blue}Response: $ Thanks and we have revised it.
>
> $\color{red}Question: $ Is using a constant resolution for $D_{rad}$ the best approach?
>
> $\color{blue}Response: $ We choose a constant resolution for the clarity and easy implementation. Uneven resolution for $D_{rad}$ is also applicable.

---

### Official Review · Reviewer_Zrsk · 2022-08-01

**Originality:** Good
**Technical Quality:** Good
**Clarity Of Presentation:** Very Good
**Impact:** 3

**Recommendation:**

Weak Reject: I recommend rejecting the paper, but will not argue for my recommendation if the majority of other reviewers have a different opinion.

**Summary:**

This paper proposes a model that projects projective images from cameras mounted on a car to a polar birds-eye-view parametersization. The polar parameterization results in a non-uniform discretization of the space which has a higher resolution near the vehicle. The results show the parameterization achieves better segmentation accuracy compared to other methods such as FIERY and CVT.

**Issues:**

- Clarify what parts of the method are leading to the improved segmentation accuracy compared to CVT and FIERY
- Explain whether the method can be used for other tasks such as estimating the road curvature.
- Fix the grammar errors


**Quality Of The Limitations Section:**

Limitations are addressed clearly

**Reviewer Expertise:**

3: The reviewer is fairly confident that the evaluation is correct

**Robotics Focus:**

Highly relevant to robotics but no hardware experiments

**Strengths And Weaknesses:**

Strengths
=========
- As far as I’m aware this is the first method that uses a polar parameterization for a birds-eye-view representation of the area around a vehicle. This parameterization is ideally suited for this application as it greatly simplifies the processing and display of information.
- The method is very simple and accounts for things like non-planar surfaces so it doesn’t make any assumptions about the curvature of the road.
- The results show that the approach achieves better performance compared to other birds-eye-view approaches such as FIERY and CVT.

Weaknesses
=========
- Apart from the detection of other car bounding boxes, there isn’t much explanation of what this parameterization and method can be used for. At the moment the functionality seems a bit limited.
- How does the approach work for obstacles that don’t have standard configurations, such as boom barriers that don’t necessarily lie on supporting surfaces?
- In terms of the evaluation, it’s not clear what contributes to the improvement in the accuracy in the segmentation task. This could either be the polar parameterization or the model itself.
- There are quite a few grammar errors that need to be fixed. Eg. “besides” is repeatedly used in a strange manner. Line 128 starts with “And”.


**Summary Of Recommendation:**

Overall, I think this type of representation is very suitable for autonomous vehicles and wheeled robots and the proposed method seems to perform well compared to existing methods. However, apart from the improved performance on bounding box estimation, it doesn't seem to show any impressive results or a significant advance in terms of theory, methods or model architectures.

---

> ### Author Response · Authors · 2022-08-28
> **Response to Reviewer Zrsk**
>
> Thanks for recognizing the novelty of our polar representation. Our method is not limited to bounding box estimation and has a wide range of use. Following are our detailed responses to your concerns.
>
> $\color{red}Question: $ Apart from the detection of other car bounding boxes, there isn’t much explanation of what this parameterization and method can be used for. At the moment the functionality seems a bit limited.
>
> $\color{blue}Response: $ We follow the evaluation of FIERY to validate the proposed PolarBEV. FIERY focuses on evaluating vehicles and we do the same for fair comparison. The functionality of PolarBEV and FIERY does not merely lie in perceiving cars. Indeed, both methods can perceive all kinds of elements in the driving scenes, including static elements (road, lane, pedestrian crossing, etc.) and dynamic elements (vehicle, pedestrian, cyclist, etc.). Extending PolarBEV to other  elements is straightfoward, i.e., just replacing the BEV ground-truth annotations and requiring no further modification about the network structure and hyper-parameters.
>
> To prove the wide range of use of our method,  we further adopt PolarBEV for road and lane segmentation. In the attached PDF file, we exhibit the predicted maps of road and lane. It shows that PolarBEV can be applied to generate rasterized HD map, which contains rich semantic information of traffic rules and serves as a fundamental component of self-driving system. Our method is not limited to bounding box estimation and has a wide range of use.
>
> As shown In Tab.2, we also conduct additional experiments on  Lyft [1] dataset to further validate our method. PolarBEV outperforms FIERY Static in terms of both performance and FPS on Lyft, showing its effectiveness.
> | Method |IoU| FPS|
> | ------ | ------ | ------ |
> | FIERY Static| 42.25 | 8 |
> | Ours | 44.69 | 22 |
> Table 2: Lyft dataset results.
>
> [1] R. Kesten, M. Usman, J. Houston, T. Pandya, K. Nadhamuni, A. Ferreira, M. Yuan, B. Low, A. Jain, P. Ondruska, S. Omari, S. Shah, A. Kulkarni, A. Kazakova, C. Tao, L. Platin- sky, W. Jiang, and V. Shet. Lyft level 5 perception dataset. 2020, 2019.
>
>
> $\color{red}Question: $ How does the approach work for obstacles that don’t have standard configurations, such as boom barriers that don’t necessarily lie on supporting surfaces?
>
> $\color{blue}Response: $ The proposed approach does NOT rely on any position assumptions of objects. The 'surface' is not a certain plane on which objects lie, but a broad concept serving for 2D-to-3D correspondence. If we project 2D images to the 3D space with the ground-truth pixel-wise depth, we get the 'ideal surface'.  In our method, we iteratively update the height of each BEV grid.  The predicted surface gradually approaches the 'ideal surface' and the 2D-to-3D correspondence is more and more accurate. Thus, our proposed approach works for boom barriers or any other elements in the driving scene.
>
> $\color{red}Question: $ In terms of the evaluation, it’s not clear what contributes to the improvement in the accuracy in the segmentation task. This could either be the polar parameterization or the model itself.
>
> $\color{blue}Response: $ Please refer to Tab. 4 and Tab. 5 in the original paper. We have provided ablation experiments to respectively validate the design of polar parameterization and height-based transformation, both of which do bring improvement in terms of inference speed and performance.
>
> We adopt the same network architecture (backbone and head design) with FIERY Static. In other words, the improvement is not from the model.
>
> $\color{red}Question: $ There are quite a few grammar errors that need to be fixed. Eg. “besides” is repeatedly used in a strange manner. Line 128 starts with “And”.
>
> $\color{blue}Response: $ Thanks for the kind advice. We have tired our best to correct grammar errors in the revised paper.

---

### Meta-Review · Area_Chair_QvR1 · 2022-08-15

**Recommendation:** Accept (Poster)
**Confidence:** 3

**Metareview:**

Dear Authors,

Thank you for submitting your manuscript to CoRL. I'm happy to inform you that your paper has been accepted. We have completed the review of your manuscript and a summary is appended below. The reviewers have advised accepting your manuscript as a poster considering your revised version and improvement of the quality of the manuscript based on the comments. Please make sure you incorporated all provided explanations and recommended editing into the final manuscript.


Regards,